# Development and Validation of a Measure of Passive Aggression Traits: The Passive Aggression Scale (PAS)

**DOI:** 10.3390/bs12080273

**Published:** 2022-08-08

**Authors:** Young-Ok Lim, Kyung-Hyun Suh

**Affiliations:** Department of Counseling Psychology, Sahmyook University, Seoul 01795, Korea

**Keywords:** scale development, passive aggression, sabotaging, criticism, ignorance, avoidance

## Abstract

Although passive aggression is known as a pathological personality trait, the concept is unclear, and there is a lack of tools to measure it comprehensively. Thus, this study developed and validated a tool for measuring passive-aggressive behaviors. Data on basic information about passive aggression traits were collected from 20 experts using open-ended questions. To verify content validity, Delphi surveys were conducted twice with five experts. Data for item analysis were collected from 123 Korean adults. Reliability and validity were analyzed using data obtained from 408 Korean adults. The three-factor model for the passive aggression scale (PAS) showed satisfactory model fits. Cronbach’s αs for inducing criticism, avoiding/ignoring, and sabotaging subscales, and the total PAS, were 0.91, 0.91, 0.92, and 0.93, respectively. The test–retest coefficient of the PAS also indicates that this tool is reliable. Analyses of the criterion-related validity revealed that the PAS was closely correlated with the scores of some scales that measure passive aggression with a single factor. In addition, the correlations between the PAS, Cook-Medley Hostility Scale, and State-Trait Anger Expression Inventory scores supported our understanding of the concept of passive aggression. This study highlights the utility of PAS as a useful and comprehensive measure of passive-aggressive behaviors to be adopted by researchers and clinicians.

## 1. Introduction

Humans, who are social beings living with others, must form healthy and smooth interpersonal relationships with others and feel a sense of stability and satisfaction in order to maintain their individual lives [1]. Most people recognize that to maintain good interpersonal relationships, they should not express aggressive words and perform actions that hurt others [2]. However, one may feel that their opinion has been ignored or treated unfairly, which generates hatred and resentment and makes one want to engage in hostile and aggressive actions [3,4]. In modern society, where violence is strongly prohibited, people rely on indirect attacks and express their hostility toward the other person verbally, timidly, or nonverbally [5]. This type of aggression is also known as indirect hostility, indirect aggression, or passive aggression.

Some people have a strong tendency toward passive aggression. In the past, it was diagnosed as a mental disorder when this tendency observed as serious. In the first edition of the Diagnostic and Statistical Manual of Mental Disorders (DSM-I), passive aggression was briefly addressed and classified together with passive-dependent personality [6], and in the text’s revised third edition (DSM-III-R), it was included in the personality disorder category as a passive-aggressive personality disorder (PAPD) [7]. The fourth version of the manual (DSM-IV), revised in 1994, describes PAPD as a disorder involving passive resistance to demands for appropriate performance and pervasive patterns of negative attitudes in various contexts [8]. However, unlike in the previous edition passive aggression here was classified as “Criteria Sets and Axes Provided for Further Study” in Appendix B rather than an Axis II personality disorder. In the manual’s latest revised edition, DSM-5, it is neither included as a personality disorder nor under “Conditions for Further Study” in the section III [9]. Although the International Classification of Diseases (ICD) of the World Health Organization classifies PAPD as “other specific personality disorder”, and argues that these personality traits could be pathological disorders for a little longer than DSM, the latest ICD-11 suggests six prominent personality traits rather than a single diagnosis of personality disorder, and passive aggression is not emphasized [10,11].

The PAPD was not included in the DSM-5 and ICD-11, because it was argued that it should be removed from the diagnostic system. One of the reasons for this change is that passive aggression is considered only one of the pathological symptoms of mental disorders [12]. Millon et al. conceptualized negativistic personality disorder including irritability, anger, discontent, and pessimism with passive aggressive traits, such as procrastination, covert obstructionism, inefficiency, and stubbornness [13]. Emotional symptoms and cognitive characteristics have been highlighted in passive aggression. Another reason the PAPD does not have an independent diagnostic code may be because it has many overlapping characteristics with other personality disorders [14].

Even though passive aggression was not included in the diagnostic system for mental disorders, it should not be dismissed. Passive aggression originates from the depths of an individual’s personality and produces pathological effects through various types of behavior. Previous studies have implied that passive aggression may be a temperament because it has genetic causal factors and develops as a result of distressing early experiences at home, including abuse and negative parenting [15,16,17]. Passive aggression may harm a person by preventing an individual from performing necessary activities. It can harm society at large as well as gives rise to negative actions such as withdrawing social support for others or ignoring others’ psychological demands and needs [18]. Passive aggression can also negatively affect mental health, causing self-harm, depression, eating disorders, and stress-related disorders, such as acute stress disorder [19,20,21,22]. Thus, we can conclude that passive aggression negatively affects both one’s own and others’ mental health.

Passive aggression may be an effort to punish dissatisfied others or restore autonomy, which can be harmful because it is intended to exert a negative influence on others [23]. In his cognitive theory, Beck also views passive aggression as a pathological reaction based on an individual’s distorted belief that they are unfairly being controlled or impeded by others [24]. Psychoanalysts might view it as a pathological pattern that occurs when an individual’s super-ego is involved in one’s ego but does not fully influence it [25]. Passive aggression has been measured as an immature self-defense mechanism that suppresses emotional conflicts and produces ineffective problem-solving behaviors [26], and, some studies have concluded that these passive-aggressive traits may be relatively stable and stimulated by internal or external stressors [27,28]. Whether passive aggression is a mental disorder, a self-defense mechanism, or something else, it requires careful attention because it negatively affects individuals and society.

A reliable and valid psychometric instrument is essential for the diagnosis, intervention, education, prevention, and study of passive aggression. The limited construct validity of the PAPD and the unsatisfactory internal consistency of its symptoms were reasons for its exclusion from the DSM [29]. Rotenstein et al. found that symptoms of the PAPD were divided into two factors but did not show clear constructs [29]. However, when analyzed with a single factor for patients with severe symptoms, the internal consistency was acceptable [30]. Hopwood et al. explored the reliability and validity of this disorder and found moderate internal consistency and satisfactory criterion-related validity. However, but they did not show a clear factorial structure [31]. An unclear factor structure could be one of the limitations of conceptualizing passive-aggressive personality.

According to the authors of this study, it is difficult to measure passive-aggressive behaviors. Controversy has continued over whether the contents of the items measuring passive-aggressive behaviors are suitable [32]. Sanchez et al. interpreted that because of many theoretical approaches (e.g., cognitive perspective, perspective as a mental disorder, defense mechanism) for passive aggression, various psychological dimensions (e.g., cognitions, emotions, personality traits), and different behaviors appear in other contexts such as in social relationships at the workplace [18]. They explain that it is difficult to utilize instruments developed to measure passive aggression in clinical settings [33], and have thus recently developed a behavior-based psychological test that can be utilized to measure passive-aggressive behaviors in clinical settings with two factors: self-directed and other-directed passive aggression [18]. This instrument is somewhat unique in that it classifies factors according to the direction of passive aggression, and it seems reasonable that self-directed passive aggression correlates with depression or somatoform symptoms. However, there are some limitations related to conceptualizing passive aggression, because the factors involved were not intended to be classified by types of passive-aggressive behaviors.

Therefore, the present study aimed to develop and validate the Passive Aggression Scale (PAS), a tool that can accurately measure passive-aggressive behaviors and clarify the concept of passive aggression. To facilitate various empirical and clinical studies on passive aggression and to provide valuable information to help prevent indirect aggression in our society, we sought to develop a scale in which the factors are classified according to types of passive-aggressive behaviors and verify its reliability and validity.

## 2. Materials and Methods

### 2.1. Participants

Twenty experts responded to open-ended questions to collect basic qualitative information on the traits of passive aggression. Among them, nine were men, and 11 were women, and their ages ranged from 31 to 60 years. Participants in the focused group interview (FGI) were college students, graduate students majoring in counseling psychology, counselors, and newspaper reporters. Among them, there were four males and eight females, with ages ranging from 20 to 52 years. Delphi surveys were conducted twice to verify content validity. The first round of a Delphi survey was conducted with two professors of Counseling Psychology and Humanities, and three counselors, and the second round of a Delphi survey was conducted with four professors of Counseling Psychology and, and a counselor.

A total of 123 Korean adults participated in the preliminary survey to analyze the items developed based on open-ended questions, FGIs, and verification of content validity. Among them, 52 were men (42.3%), and 71 were women (57.7%), with ages ranging from 18 to 58 years with an average age of 41.59 ± 10.80 years. Twenty-seven (22.0%) participants were in their 20s or younger, 14 (11.4%) were in their 30s, 62 (50.4%) were in their 40s, and 20 (16.3%) were in their 50s and above.

This study’s main analyses, the confirmatory factor analysis (CFA) and analysis of criterion-related validity, included 408 Korean adults as participants. Among the participants, 172 were men (42.2%), and 236 were women (57.8%), and their ages ranged from 18 to 66 years with an average age of 37.52 ± 12.68 years. A total of 143 (35.0%) participants were in their 20s or below, 60 (14.7%) in their 30s, 117 (28.7%) in their 40s, and 88 (21.6%) in their 50s and above. Among them, 89 (35 males and 54 females) participated in the preliminary survey and were subjected to analysis for test–retest reliability.

### 2.2. Data Gathering Procedure

Data for the preliminary survey and the main analyses were collected using a questionnaire posted on Google. The recruitment of respondents took place through the University’s internet bulletin board and on social networking services (SNSs), where adults participated. We attempted to match the ratio of gender and age groups, although we did not force this distribution. The participants were asked to submit their questionnaires when all items were answered.

This study was approved by the Institutional Review Board (IRB approval number: 2-1040781-A-N-012020083HR) before data collection, and all procedures were conducted ethically. The elements of written informed consent for the online survey were presented to participants online. Participants were informed that even those who agreed to answer the online survey could withdraw at any time while responding to the questionnaire.

### 2.3. Instrument

In addition to the preliminary passive aggression items developed in this study, several other instruments were used to examine criterion-related validity.

#### 2.3.1. Preliminary Passive Aggression Items

Ninety-five items measuring passive aggression were developed, using open-ended questionnaires and the FGI. Each item used a six-point Likert scale (1 = *not at all true*, 2 = *not quite true*, 3 = *little not true*, 4 = *little true*, 5 = *quite true*, and 6 = *very true*). In addition, 81 preliminary items were selected through two rounds of content validity verification by professors and psychologists, and through facial validity verification with five graduate students majoring in counseling psychology and five laypeople. Among the 81 preliminary items, five items with response values of less than 1.5, which were too far from the average, nine items with standard deviations of less than 0.9 [34], seven items with skewness of 2 or more deviating from the normal distribution, and four items showing correlation coefficients of more than 0.80 with a certain item were excluded [35]. However, some items overlapped with these conditions; thus, 11 items were removed from this primary item analysis. Finally, we performed exploratory factor analysis (EFA) with 70 items.

The root mean square error of approximation (RMSEA), which is less sensitive to sample size and considers simplicity, was used to ascertain the number of factors of the PAS to retain based on 70 items. An RMSEA difference greater than 0.01 should be retained for rotation [36]. Since the RMSEA was 0.073 (*χ*^2^ = 7147.89, *df* = 2276) when the number of factors was two, 0.063 (*χ*^2^ = 5789.90, *df* = 2208) when the number of factors was three, and 0.060 (*χ*^2^ = 5245.82, *df* = 2141) when the number of factors was four, we decided the number of factors to be three. We fixed the number of factors to three and performed the EFA with maximum likelihood and oblique rotation for the primarily arranged 70 items. Items with a communality of less than 0.50 and items loaded on two factors with a difference in values of factor loading of less than 0.10 were removed. Subsequently, we performed the EFAs in the same way as the remaining items and removed the items under the same conditions. Finally, 17 items were included in Factor 1, 14 in Factor 2, and seven in Factor 3. To arrange the same number of items for each factor, seven items with high factor loading among the items in Factor 1 and Factor 2 were selected along with the number of items in Factor 3, and we performed an EFA with these 21 items.

#### 2.3.2. Ewha Defense Mechanism Test (EDMT)

The EDMT is a widely used defense mechanism test in Korea that was developed, validated, and standardized by Rhee et al. [37]. Items of this scale were developed based on traditional Korean proverbs, based on the premise that defense mechanisms are major factors in understanding an individual’s personality and adaptive ability. This scale consists of 200 items with 20 subscales; although we only used the subscale of passive-aggressive defense mechanisms; it uses a five-point Likert scale ranging from 1 (*not at all true*) to 5 (*very true*). In this study, the internal consistency of the 10 items (Cronbach’s α) was 0.83.

#### 2.3.3. Defense Style Questionnaire (DSQ)

We used the Korean version of the Defense Style Questionnaire (K-DSQ), originally developed by Bond et al., and validated by Cho for Koreans [38,39]. The DSQ consists of 16 defense mechanisms, which are it is divided into four factors in the factor analysis (28 items: immature defense mechanisms, 15 items: adaptive defense mechanisms, 16 items: self-suppressive defense mechanisms, and six items: conflict-avoiding defense mechanisms). In this study, we used only five items from the immature defense mechanism subscales that measure passive aggression. The DSQ uses a five-point Likert scale ranging from 1 (*strongly disagree*) to 5 (*strongly agree*). Cronbach’s α was 0.80 in this study.

#### 2.3.4. Minnesota Multiphasic Personality Inventory Personality Disorder Scale (MPDS)

We used 24 items of the MPDS, which measures the trait of PAPD. The MPDS was created by reconstructing with some items of the Minnesota Multiphasic Personality Inventory (MMPI) by Park et al. to measure personality disorders [40]. Each item was either true or false, forced-choice item, and one item was a reversed item. Cronbach’s α for the 24 items was 0.82 in this study.

#### 2.3.5. Buss–Durkee Hostility Inventory (BDHI)

The BDHI originally consisted of 75 items with eight subscales: assault, indirect hostility, irritability, negativism, resentment, suspicion, verbal hostility, and guilt [41]. We used only nine items from the indirect aggression subscale of the Korean version of the BDHI translated by Hong and Roh [42]. Each item was a forced-choice item, and the Cronbach’s α was 0.60 in this study.

#### 2.3.6. Cook–Medley Hostility Scale (Ho)

‘Ho’ was developed by Cook and Medley and comprises 64 items, including 27 items and 37 filler items [43]. Cook and Medley attempted to select items from the MMPI to develop a hostility scale to measure an individual’s disposition to cynicism and aggression derived from chronic hatred [44]. Ho has three subscales: cynicism (cynical hostility), hostile affect, and aggressive responses. We used the Korean version of the scale developed by Chon and Kim [45] and only 13 items of the cynicism subscale. Each item was a forced-choice item, and the Cronbach’s α for these items was 0.83.

#### 2.3.7. State–Trait Anger Expression Inventory (STAXI)

The STAXI was developed by Spielberger et al. [46], and comprises 44 items to assess state anger (10 items), trait anger (10 items), and three anger expression subscales: anger-in (8 items), anger-out (8 items), and anger control (8 items). In this study, 24 items from the anger expression subscales, translated into Korean by Chon et al., were used [47]. STAXI uses a four-point Likert scale ranging from 1 (*almost never*) to 4 (*almost always*), and in this study, the Cronbach’s αs of anger−in, anger−out, and anger control were 0.81, 0.86, and 0.88, respectively

### 2.4. Statistical Analysis

IBM SPSS (Statistical Package for the Social Sciences) Statistics for Windows 26.0 and Analysis of Moment Structure (AMOS) 23.0 were used for all statistical analyses. SPSS was used to calculate the mean, standard deviation, skewness, and kurtosis of the data, which were checked for a normal distribution for parametric statistical analyses. Pearson-product moment correlational analysis and EFAs were performed using SPSS, and CFAs were performed using AMOS.

The goodness-of-fit of the CFAs was assessed using the Tucker–Lewis index (TLI), comparative fit index (CFI), standardized root mean square residual (SRMR), and RMSEA. Generally, RMSEA and SRMR smaller than 0.08 suggest a satisfactory model fit, and TLI and CFI, larger than 0.90, suggest a good model fit [48]. Composite reliability (CR) and average variance extracted (AVE) were calculated to assess convergent validity. A CR larger than 0.70 and an AVE larger than 0.50 suggest good convergent validity [49].

## 3. Results

### 3.1. EFA of the PAS

To measure the suitability of these data for an EFA, the Kaiser–Meyer–Olkin (KMO) was checked; it was 0.93 (>0.80) for 21 items, indicating that it was a suitable sample for factor analysis. As shown in Table 1, the EFA revealed that three factors accounted for approximately 67.8% of the total variance (eigenvalues > 1.0: 9.08, 3.19, and 1.87). Factor 1, in which seven items described “inducing criticism”, accounted for 43.3% of the total variance, and the factor loadings ranged from 0.528 to 0.873. Factor 2, in which seven items described “avoiding or ignoring,” accounted for an additional 15.2% of the total variance, and the factor loadings ranged from 0.672 to 0.914. Finally, Factor 3, in which the seven items described “sabotaging,” accounted for an additional 9.4% of the total variance, and the factor loadings ranged from −0.591 to −0.861. In addition, the absolute values for skewness of the PAS subscales ranged from 0.08 to 1.03 and the absolute values for kurtosis ranged from 0.45 to 0.95, suggesting that they were close to the normal distribution.

### 3.2. CFA of the PAS

To measure the suitability of these data for EFA, the KMO was checked; it was 0.93 (>0.80) for 21 items indicating that it was a suitable sample for factor analysis. As shown in Table 1, the EFA revealed that three factors accounted for approximately 67.8% of the total variance (eigenvalues > 1.0: 9.08, 3.19, and 1.87).

CFA was performed using the PAS. The *χ*^2^ value of the three-factor model was 660.26 (*df* = 186, *p* < 0.001), and the goodness-of-fit index was TLI = 0.909, CFI = 0.919, SRMR = 0.065, and RMSEA = 0.079 (CI: 0.073 to 0.086). RMSEA and SRMR (below 0.08) were within the range of indices for good model conditions, and TLI and CFI (above 0.90) were satisfactory (Table 2).

Following the modification indices (MI), we examined the goodness–of–fit after allowing two error covariances for items 12 and 13 and items 16 and 19. The adjusted three-factor model was an improved fit. The levels of TLI and CFI were 0.923 and 0.932 respectively, and SRMR and RMSEA also improved (SRMR = 0.063 and RMSEA = 0.073 [CI: 0.066–0.080]). 

The standardized regression weights (SRWs) in this CFA of the PAS, allowing for the two-error covariances, are shown in Figure 1. The SRWs for the inducing criticism subscale ranged from 0.70 to 0.86. In addition, the SRWs for the avoiding/ignoring subscale ranged from 0.71 to 0.83, while the SRWs for the sabotaging subscale ranged from 0.73 to 0.86. In this model, the estimated correlation between the inducing criticism and avoiding/ignoring subscales was 0.35, the estimated correlation between the inducing criticism and sabotaging subscales was 0.61, and the estimated correlation between the avoiding/ignoring and sabotaging subscales was 0.54.

In Table 3, the CRs of the inducing criticism, avoiding/ignoring, and sabotaging subscales of the PAS were 0.92, 0.91, and 0.92, respectively. The CRs of all subscales were above 0.70. In addition, the AVEs of subscales were also satisfactory because, in each case, the AVE was above 0.50 (0.61 for inducing criticism, 0.58 for avoiding/ignoring, and 0.63 for sabotaging).

Cronbach’s *α*s, which indicate the degree of internal consistency for inducing criticism, avoiding/ignoring, and sabotaging subscales, were 0.91, 0.91, and 0.92, respectively, while Cronbach’s *α* for the 21-item PAS was 0.93. The test–retest coefficients for inducing criticism, avoiding/ignoring, and sabotaging subscales were 0.63, 0.76, and 0.72, respectively. Tests and retests were conducted for 89 people at intervals of four weeks, and the test–retest coefficient for the total score of the 21-item PAS was 0.77.

### 3.3. Criterion-Related Validity of the PAS

To identify the concurrent and predictive validity of the criterion-related validity, we analyzed how the PAS correlated with the passive aggression subscale of EDMT (EDMT-PA), passive aggression items of DSQ (DSQ-PA), passive aggression subscale of MDPA (MDPA-PA), the indirect aggression subscale of BDHI (BDHI-IA), the cynicism subscale of Ho, anger-in, anger-out, and anger control (Table 4). None of the absolute values for skewness and kurtosis of the EDMT-PA, DSQ-PA, MDPA-PA, BDHI-IA, cynicism, anger-in, anger-out, and anger control exceeded 1.0, thus satisfying the conditions for conducting parametric statistical analyses.

The correlational analysis revealed that the total PAS scores were strongly correlated with the EDMT-PA (*r* = 0.70, *p* < 0.001) and DSQ-PA (*r* = 0.65, *p* < 0.001), which measure passive-aggressive trait in Korea. In particular, the sabotaging subscale shared a 38.4% (*r* = 0.62) variance with the EDMT-PA and DSQ-PA scores. PAS showed moderately high correlation coefficients with MPDS-PA (*r* = 0.31, *p* < 0.001) and BDHI-IA (*r* = 0.49, *p* < 0.001), which measured the passive aggression trait and indirect aggression with the forced-choice items.

In addition, PAS scores were positively correlated with cynicism (*r* = 0.40, *p* < 0.001) and anger-in (*r* = 0.40, *p* < 0.001). The PAS shared 16.0% variance with cynicism and anger-in, while it shared 5.8% variance with anger-out (*r* = 0.24, *p* < 0.001) and 2.3% variance with anger-control (*r* = −0.15, *p* < 0.01).

## 4. Discussion

This study developed a behavior-based and self-reported PAS, and examined the reliability and validity of the PAS to confirm that it is a useful tool for measuring passive-aggressive behaviors or personality in clinical and academic settings. Based on the results of this study, we realized that the newly developed PAS compensated for the limitations of existing measures of passive aggression to a certain extent. The implications of these findings are discussed below.

The 21-item PAS developed in this study showed a stable factorial structure with the following subscales: inducing criticism, avoiding/ignoring, and sabotaging. In the CFA, the model fit indices of the PAS’s factorial structure with three factors allowing for two error covariances were excellent. The factorial structure of the PAS with three factors that did not allow for any error covariance was satisfactory, and these three factors accounted for more than 60% of the total variance on this scale [50]. In summary, this study demonstrated satisfactory construct validity for the three-factor model of the PAS.

The CRs of all three PAS subscales were above 0.90 and thus, may be classified as excellent The AVEs of the subscales were also satisfactory, at approximately 0.60 [49]. These results indicate that the three-factor model of PAS has convergent validity. Consequently, this evidence supports the construct validity of the three-factor PAS model. If so, the results of this study can be considered a development that goes far beyond measuring passive aggression with only one factor [37,38,39,40,41], and one that complements the limitations of studies developing scales with an unclear factorial structure for measuring passive aggression [31,32].

After conducting a factor analysis of the various items developed, we concluded that this three-factor structure was logically valid. For example, in the process of developing items, it was logically reasonable that one of the items (“*I pretend I am the victim to make someone I dislike or find uncomfortable to give them a hard time*”), in which we assumed a factor of “pretending to be a play a victim” or “victim cosplay,” was included in the factor of “inducing criticism”. Notably, the items meaning “avoiding” and “ignoring” converged into one factor. This result suggests that performing avoidance and neglect toward a person someone dislikes is similar to exhibiting passive-aggressive behaviors against that person.

Internal consistency of the PAS was excellent, because Cronbach’s *α*s for all PAS subscales were above 0.90 and Cronbach’s *α* for all items of the PAS was 0.93 [51]. This means that the PAS noticeably compensated for the limitations of existing measures of passive-aggressive behaviors with unsatisfactory internal consistency [29,30]. In addition, this internal consistency was better than the internal consistencies of the scales used in this study to measure passive aggression to verify criterion-related validity [37,39,40]. The four-week interval test–retest reliability of the PAS was also satisfactory, because the test–retest coefficient for the total PAS score was 0.77. This means that the 21-item PAS has excellent reliability. In other words, this scale is reliable for measuring passive aggression. In addition, some studies have found that passive aggression develops from early childhood [27,28], which suggests that passive aggression is a stable personality trait that does not change easily over time unless approached strategically and purposively.

The PAS also demonstrated satisfactory criterion-related validity, sharing 49.0% (*r* = 0.70) and 42.3% (*r* = 0.65) of variances with the EDMT-PA and DSQ-PA, respectively, which measure Koreans’ passive aggression with a single factor as a defense mechanism. This result indicates that the PAS would be useful in measuring individuals’ passive aggression with specific factors, such as inducing criticism, avoiding/ignoring, and sabotaging. The EDMT-PA and DSQ-PA shared more variance with the sabotaging subscale than with the inducing criticism or avoiding/ignoring subscales. Rather than inducing criticism or individuals avoiding and ignoring people they dislike or hate, sabotaging them is a more direct passive-aggressive behavior that can cause more damage to their opponents. Therefore, the results of this study suggest that the EDMT-PA and DSQ-PA measure passive-aggressive behaviors that can cause slightly more damage to opponents.

PAS scores were also correlated with MPDS-PA and BDHI-IA. It was found to share relatively little variance with MPDS-PA, which measures passive aggression. We assumed that this is because the MPDS-PA selected items related to passive aggression from the MMPI, rather than those originally developed to measure passive aggression. However, the MPDS-PA and BDHI-IA shared more variance with the avoiding/ignoring subscale than with the sabotaging or inducing criticism subscales. This is because, as mentioned earlier, MPDS-PA items were originally not developed to measure passive aggression, and the BDHI-IA would have been more related to avoiding or ignoring rather than sabotaging or inducing criticism, because it is a measure of indirect aggression. These results provide evidence that the PAS is a useful tool for comprehensively measuring indirect passive-aggressive behaviors.

The correlations between PAS and Ho or STAXI scores also support the validity of the former. PAS was positively correlated with cynical hostility, and all PAS subscales, inducing criticism, avoiding/ignoring, and sabotaging were significantly correlated with cynicism. This result reiterates that hostility exists at the base of all kinds of passive-aggressive behaviors. Although PAS was positively correlated with both anger-in and anger-out, the result that showed more variance shared with anger-in than anger-out is evidences that the PAS measures indirect aggressive behaviors rather than direct aggressive behaviors. In addition, the negative correlation between PAS and anger control suggests that passive-aggressive behaviors were caused by failure to fully control anger or hostility.

There are also implications for educational communities at the psychosocial level. Since the PAS has been validated as a useful instrument for measuring individuals’ traits of passive aggression, it could be an elementary tool for treating the negative effects of passive aggression in interpersonal relationships, not only clinically but also educationally. For example, Cebollero-Salinas et al. suggested the necessity of preventive education for aggressive traits developed during childhood because problematic aggression such as cyber gossip can appear in the cyber environment [52]. In addition, Bisquerra et al. emphasized the importance of cultivating moral emotions [53], and it would be useful to cultivate moral emotions or the emotional intelligence of children or adolescents based on the factors involved with passive aggression, as shown in this study. Furthermore, passive-aggressive behavior is known to ruin social relationships [18,54]; and the PAS can be used to assist people in initiating or recovering healthy interpersonal relationships. Furthermore, we expect that the PAS and the sub-concepts of passive aggression derived from the scale-development process could be useful when applied to educational policies.

## 5. Conclusions

Although we found evidence that the PAS developed in this study can be a useful tool for measuring passive aggression, some limitations should be considered when interpreting this study’s results. First, the sample of this study was not perfectly representative of people globally, because data collection was conducted online and from stratified samples in Korea. Second, the PAS developed in this study consisted only of behavior-based items and did not include thoughts or emotions related to passive aggression. Third, PAS items do not include specific social environments or job situations, suggesting that the PAS can measure passive-aggressive behaviors comprehensively in all situations, which may not be the case. Fourth, those who participated in the survey for the main analysis and those who participated in the preliminary survey were not completely independent samples. Finally, the PAS may not be able to measure all the dimensions of passive aggression (e.g., self-directed passive aggression). Further studies must conduct scale-development research to overcome these limitations.

This study developed the PAS and demonstrated that it could be a useful tool for measuring passive aggression. The PAS with the three-factor model showed good factorial structure, excellent model fit, and excellent convergent validity. The reliability of the PAS was also satisfactory because the estimated internal consistency was high, and the test-retest coefficient was moderate. This scale has shown excellent concurrent and predictive validity, lending credence to its utility as a comprehensive measure of passive-aggressive behavior that can be used for research, education, and clinical purposes.

## Figures and Tables

**Figure 1 behavsci-12-00273-f001:**
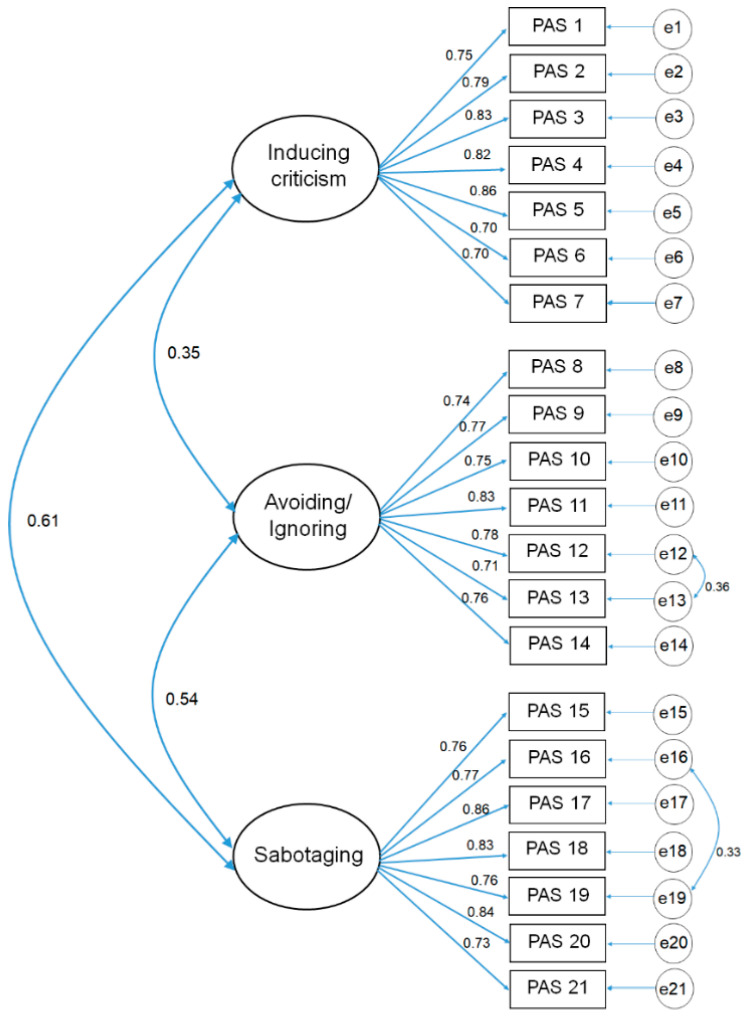
Construct model of the PAS, three-factor model.

**Table 1 behavsci-12-00273-t001:** Factor structure matrix of the passive aggression scale (PAS).

NO	Items	Factor Loadings			*h* ^2^
		1	2	3	
1.	When I talk about someone I dislike or find uncomfortable, I pretend to praise their strengths but also drop hints about their weaknesses. 나는 싫어하는 사람이나 불편한 사람의 강점을 이야기하는 척하면서 그의 약점을 드러내곤 한다.	0.823			0.576
2.	I tattle on mistakes made by someone I don’t like or find uncomfortable to a higher authority to ruin their reputation. 나는 싫어하는 사람이나 불편한 사람의 이미지를 깎아 내리기 위해 영향력 있는 사람에게 고의로 그의 실수를 언급한다.	0.725			0.631
3.	I intentionally reveal embarrassing events or the dark pasts of someone I dislike or find uncomfortable in public. 나는 싫어하는 사람이나 불편한 사람의 좋지 못한 과거사를 의도적으로 공개하곤 한다.	0.845			0.661
4.	I ask someone I don’t like or find uncomfortable questions they can’t answer in front of others to make them uncomfortable. 나는 싫어하는 사람이나 불편한 사람에게 무안을 주기 위해 그가 모르는 것을 대중 앞에서 질문한다.	0.813			0.641
5.	I mock someone I don’t like or find uncomfortable by being sarcastic and pretending it’s just a joke. 나는 싫어하는 사람이나 불편한 사람에게 농담을 가장해 비꼬는 표현을 사용하곤 한다.	0.873			0.707
6.	When I have something I want to say about someone I dislike or find uncomfortable, I talk about it with others in plain sight of them. 나는 싫어하는 사람이나 불편한 사람에게 하고 싶은 말을 그 사람이 있는 상황에서 의도적으로 다른 사람에게 말한다.	0.608			0.508
7.	I pretend I am the victim to give someone I dislike or find uncomfortable a hard time. 나는 싫어하는 사람이나 불편한 사람을 곤경에 빠뜨리기 위해 내가 피해자인 척한다.	0.528			0.546
8.	I purposefully avoid eye contact with someone I don’t like or find uncomfortable. 나는 싫어하는 사람이나 불편한 사람에게 일부러 눈길을 주지 않는다.		0.802		0.564
9.	When I meet someone I dislike or find uncomfortable, I try to get away from them intentionally. 나는 싫어하는 사람이나 불편한 사람을 마주칠 때 고의로 자리를 피한다.		0.703		0.583
10.	I cut ties with someone I dislike or find uncomfortable even though I know they want to get in touch with me and find out how I am doing. 나는 싫어하는 사람이나 불편한 사람이 궁금해하는 것을 알면서도 의도적으로 연락을 끊는다.		0.667		0.544
11.	When someone I dislike or find uncomfortable tries to connect with me by phone, I deliberately choose to ignore them. 나는 싫어하는 사람이나 불편한 사람의 연락을 고의로 무시하곤 한다.		0.761		0.628
12.	I give someone I dislike or find uncomfortable the silent treatment. 나는 싫어하는 사람이나 불편한 사람에게 침묵으로 일관한다.		0.914		0.699
13.	When someone on SNS I dislike or find uncomfortable asks me a question, I pretend I never saw the question in the first place. 나는 SNS에서 싫어하는 사람이나 불편한 사람이 답을 요구하는 것에 대하여 못 본 척한다.		0.735		0.586
14.	I have a cold and dismissive attitude toward someone I dislike or find uncomfortable. 나는 싫어하는 사람이나 불편한 사람의 중요한 일에 냉소적인 태도를 보이곤 한다.		0.672		0.577
15.	I deliberately delay someone I dislike or find uncomfortable to give them a hard time. 나는 싫어하는 사람이나 불편한 사람의 일을 미적거리면서 애를 태운다.			−0.683	0.590
16.	I pretend to help someone I dislike or find uncomfortable but sabotage their work behind their back. 나는 싫어하는 사람이나 불편한 사람의 일을 돕는 척하나 결국 일을 망쳐 버린다.			−0.856	0.677
17.	When I work with someone I dislike or find uncomfortable, I intentionally don’t do my share of the work and end up penalizing them. 나는 싫어하는 사람이나 불편한 사람과 일을 함께 할 때 내가 할 일을 고의로 하지 않아 손해를 끼치곤 한다.			−0.861	0.721
18.	I come up with excuses and say things like: “I forgot” to someone I dislike or find uncomfortable. 나는 싫어하는 사람이나 불편한 사람의 일을 깜박 잊어버렸다고 변명한다.			−0.796	0.667
19.	I deliberately procrastinate when someone I dislike or find uncomfortable asks me to do something. 나는 싫어하는 사람이나 불편한 사람이 시키는 일은 일부러 지연시킨다.			−0.789	0.666
20.	When someone I dislike or find uncomfortable asks me for a favor, I don’t give it my all and do a sloppy job. 나는 싫어하는 사람이나 불편한 사람의 부탁을 의도적으로 허술하게 수행한다.			−0.740	0.708
21.	When someone I dislike or find uncomfortable asks me to do something, I don’t do it properly and come up with excuses like: “I didn’t know it was important.” 나는 싫어하는 사람이나 불편한 사람이 시킨 일을 고의로 아무렇게 진행한 후, 나중에 중요한지 몰랐다고 변명한다.			−0.592	0.567
Eigenvalues		9.08	3.19	1.97	
% Variance		43.25	15.19	9.36	67.80
Skewness		1.03	−0.08	0.99	0.38
Kurtosis		0.67	−0.46	0.95	−0.10

**Table 2 behavsci-12-00273-t002:** Goodness of fit of confirmatory factor analyses for the PAS.

Model	χ^2^	*df*	TLI	CFI	SRMR	RMSEA (90% Confidence Interval)
Three-factor model	660.26 ***	186	0.909	0.919	0.065	0.079 (0.073~0.086)
Three-factor model with two-error covariance	580.86 ***	184	0.923	0.932	0.063	0.073 (0.066~0.080)

*** *p* < 0.001.

**Table 3 behavsci-12-00273-t003:** Construct validity and reliability of PAS.

Constructs	Item Number	SRW	SE	CR	AVE	Alpha	Test–Retest Coefficients
Inducing criticism	PAS 1	0.75	−	0.92	0.61	0.91	0.63
	PAS 2	0.79	0.07				
	PAS 3	0.83	0.07				
	PAS 4	0.82	0.06				
	PAS 5	0.86	0.06				
	PAS 6	0.70	0.07				
	PAS 7	0.70	0.07				
Avoiding/ Ignoring	PAS 8	0.74	−	0.91	0.58	0.91	0.76
	PAS 9	0.76	0.06				
	PAS 10	0.75	0.07				
	PAS 11	0.83	0.07				
	PAS 12	0.78	0.07				
	PAS 13	0.71	0.06				
	PAS 14	0.76	0.06				
Sabotaging	PAS 15	0.76	−	0.92	0.63	0.92	0.72
	PAS 16	0.77	0.05				
	PAS 17	0.86	0.06				
	PAS 18	0.83	0.06				
	PAS 19	0.76	0.05				
	PAS 20	0.85	0.06				
	PAS 21	0.73	0.05				
Total						0.93	0.77

**Table 4 behavsci-12-00273-t004:** Correlational matrix of EDMT, MPDS, DSQ, BDHI, cynicism, anger-in, anger-out, anger control, and PAS (*n* = 408).

Scale	EDMT-PA	DSQ-PA	MPDS-PA	BDHI-IA	Cynicism	Anger-In	Anger-Out	Anger-Con
Inducing criticism	0.54 ***	0.51 ***	0.23 ***	0.33 ***	0.33 ***	0.22 ***	0.22 ***	−0.17 ***
Avoiding/Ignoring	0.53 ***	0.45 ***	0.32 ***	0.47 ***	0.31 ***	0.43 ***	0.23 ***	−0.05
Sabotaging	0.62 ***	0.62 ***	0.19 ***	0.35 ***	0.28 ***	0.23 ***	0.11 *	−0.15 **
PAS	0.70 ***	0.65 ***	0.31 ***	0.49 ***	0.40 ***	0.40 ***	0.24 ***	−0.15 **
Skewness	0.28	0.31	0.65	−0.24	0.31	0.64	0.69	−0.33
Kurtosis	−0.15	−0.70	0.03	−0.41	0.01	0.16	0.85	0.05

* *p* < 0.05, ** *p* < 0.01, *** *p* < 0.001.

## Data Availability

The datasets analyzed in this study are available from the corresponding author upon reasonable request.

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
