# Peer review of "Development and Validation of a Measure of Passive Aggression Traits: The Passive Aggression Scale (PAS)"

_behavsci, 2022, doi:10.3390/bs12080273_

Round 1

Reviewer 1 Report

We value the changes introduced by the authors. We believe that greater implications can be introduced at the psycho-socio-educational level for educational communities. We miss the contributions and implications for educational policies. We value, once again, the contributions made.

Author Response

Response to Reviewer 1 Comments

Point 1. We value the changes introduced by the authors. We believe that greater implications can be introduced at the psycho-socio-educational level for educational communities. We miss the contributions and implications for educational policies. We value, once again, the contributions made.

Response 1: Thank you very much for your comment. We included the following two sentences in the discussion regarding what you commented.

There are also implications at the psycho-socio-educational level for educational communities. (Line 427-428)

Furthermore, we expect that PAS and the sub-concepts of passive aggression derived from scale development process can be used to apply educational policy. (Line 438-440)

We will do English proofreading if this manuscript will be accepted.

Reviewer 2 Report

The authors have significantly worked on the manuscript, and in the updated version, they have made significant changes. No more comments

Author Response

Response to Reviewer 2 Comments

Thank you very much for your evaluation and cheering us that this study is valuable. The revised parts were marked in red, and we included the page and line of the revised part.

Point 1. The authors have significantly worked on the manuscript, and in the updated version, they have made significant changes. No more comments

Response 1: Thank you for your good evaluation. We will do English proofreading if this manuscript will be accepted.

This manuscript is a resubmission of an earlier submission. The following is a list of the peer review reports and author responses from that submission.

Round 1

Reviewer 1 Report

An interest in the subject is evident, however, the need to reinforce the selected samples is questioned,
as well as the relevance of incorporating transcultural aspects in the future that may have their weight and
influence on the usefulness of the instrument.
The approach of the tool is interesting, however, it is suggested that it be taken into account not so much
a pathological view of behavior, but rather a positive perspective. This positive perspective could be based
on the emotional competencies that have been studied with authors such as Bisquerra, as well as from the
perspective of e-competencies in online communicative environments with the reference of Cebollero Salinas
(For example, DOI: 10.1016/j.chb.2022.107230).

It is suggested to reinforce the psychological, social and educational implications in order to make the scale
more useful among the different professionals.

We hope that some of the comments and contributions to the presented paper will be useful.
The time invested in its analysis, carried out in an agile way, we hope will be fruitful to improve the scientific contribution.

Finally, I state the value of what has been contributed by the authors, making the pertinent nuances with a constructive spirit.

In summary: 

The typology of work is valued since it is necessary to implement scales on these topics, however, transfer and utility must be reinforced not only for researchers but also for the educational community.

Statistical data at a descriptive level are analyzed in a simple way.

On the other hand, for the object of research, it is suggested to expand with qualitative methodologies that allow a deeper and more holistic understanding about the subject.

The references provided are valued, although it is suggested that they be reinforced with more up-to-date contributions.

The conclusions must be reinforced in order to delve deeper into the socio-educational implications of the scale. It is suggested to highlight and reinforce the statistical assessments of the experts on the validation of the scale.

The limitations indicated are clear and evident. However, it is suggested to give some kind of response to these limitations to improve the quality of the work.

Reviewer 2 Report

In this study the author have evaluated the development and validation of a measure of passive aggression traits : the PAS. In this study, the authors have performed very strong statistical analysis, the authors should be applauded for that, however the presentation of the manuscript is confusing at times. Here are some of the comments.

  1. Introduction is well written, but as the authors have discussed importance of passive aggressive scale is still  questionable as it has been removed from DSM-V. So, value of this topic is not that importance for the clinical practice.
  2. In the method section please provide detail on experts. In the last sentence of 2.3.1 it is mentioned that 70 items were selected, however it is not clear as based on table 1 there were 21 items.
  3. Throughout the manuscript so many statistical terminologies have been used however it is difficult to interpret for clinicians, and result-discussion section should be easy to interpret.
  4. Based on Table 4 the correlation is not strong enough despite being statistically significant, which is not discussed in the discussion section.

Other than this strenght of this manuscript is strong statistical analysis, and well written introduction section. Weakness is result and discussion section has too many statistical jargons and not easy to understand for clinicians. In addition, methodology regarding participants for this scale is confusing and not easily understandable. Major weakness is even if we apply this scale in clinical practice it may not have strong clinical significance, which was one of the reason passive-aggresive personality has been removed from recent DSMs.